# Impacts of a DUF2207 Family Protein on *Streptococcus mutans* Stress Tolerance Responses and Biofilm Formation

**DOI:** 10.3390/microorganisms11081982

**Published:** 2023-08-01

**Authors:** Xiaochang Huang, Camile G. Laird, Paul P. Riley, Zezhang Tom Wen

**Affiliations:** 1Department of Oral and Craniofacial Biology, School of Dentistry, Louisiana State University Health Sciences Center, New Orleans, LA 70112, USA; huangxchjxlab@outlook.com (X.H.); rrile1@lsuhsc.edu (P.P.R.); 2Department of Microbiology, Immunology and Parasitology, School of Medicine, Louisiana State University Health Sciences Center, New Orleans, LA 70112, USA

**Keywords:** *Streptococcus mutans*, DUF2207 family of proteins, UppP, biofilms, stress tolerance response, cell envelope biogenesis, bacitracin resistance

## Abstract

Locus SMU.243 in *Streptococcus mutans* was annotated as a member of the DUF2207 family proteins highly conserved in all bacteria but with unknown function. To investigate its role in *S. mutans* physiology, a SMU.243-deficient mutant was constructed using allelic exchange mutagenesis, and the impacts of SMU.243 deletion on bacterial growth, stress tolerance response, and biofilm formation were analyzed. Compared to the wild-type UA159, *S. mutans* lacking SMU.243 displayed a reduced growth rate and a reduced overnight culture density (*p <* 0.01) when grown at low pH and in the presence of methyl viologen. Relative to the parent strain, the deficient mutant also had a reduced survival rate following incubation in a buffer of pH 2.8 (*p* < 0.01) and in a buffer containing hydrogen peroxide at 58 mM after 60 min (*p* < 0.001) and had a reduced capacity in biofilm formation especially in the presence of sucrose (*p* < 0.01). To study any ensuing functional/phenotypical links between SMU.243 and *uppP*, which is located immediately downstream of SMU.243 and encodes an undecaprenyl pyrophosphate phosphatase involved in recycling of carrier lipid undecaprenyl phosphate, a *uppP* deficient mutant was generated using allelic exchange mutagenesis. Unlike the SMU.243 mutant, deletion of *uppP* affected cell envelope biogenesis and caused major increases in susceptibility to bacitracin. In addition, two variant morphological mutants, one forming rough colonies and the other forming mucoid, smooth colonies, also emerged following the deletion of *uppP*. The results suggest that the SMU.243-encoded protein of the DUF2207 family in *S. mutans* plays an important role in stress tolerance response and biofilm formation, but unlike the downstream *uppP*, does not seem to be involved in cell envelope biogenesis, although the exact roles in *S. mutans*’ physiology awaits further investigation.

## 1. Introduction

*Streptococcus mutans*, a major etiological agent in dental caries, lives primarily in the plaque biofilms on the tooth surface, a highly diverse and dense microbiota [1]. It possesses multiple mechanisms to colonize the tooth surface which include the high-affinity, multi-function adhesin P1 (aka SpaP, AgI/II, and PAc) that binds to the salivary glycoprotein GP340, a primary pathway of the bacterial colonization in the absence of sucrose [2]. P1 also forms amyloid fibrils which along with the extracellular deoxyribonucleic acids (eDNA) and the adhesive exopolysaccharides of the glycosyltransferases (Gtfs) constitute the extracellular biofilm matrices [3]. *S. mutans* possesses at least three glucosyltransferases (GtfB, C&D) that utilizes the extracellular polysaccharides, including the adhesive water insoluble glucans from mostly GtfB when growing in sucrose, starch, and some other readily metabolizable sugars, which is well known as the major contributor to *S. mutans*’ adherence and accumulation on the tooth surface and its ability to cause carious lesions [4]. As a major cariogenic bacterium, *S. mutans* also plays a key role in the establishment of many other cariogenic microorganisms, such as Candida albicans and Lactobacillus spp. and the development of cariogenic biofilms [5,6]. In a rat caries model, co-infection of *S. mutans* and Candida albicans significantly increased C. albicans in the plaque microbiota and enhances plaque biofilm virulence leading to aggressive onset and rampant carious lesions [7].

As the natural habitat of *S. mutans* and other microorganisms, the oral cavity is an environment that is featured with frequent and sometimes dramatic fluctuations in conditions such as pH, nutrition sources and availability, and oxygen tension. Oral care products, such as toothpastes and mouth rinses also contain a variety of antimicrobial agents, including hydrogen peroxide, sodium lauryl sulfate, and chlorhexidine. Many microorganisms in the diverse and dense microbial community can also produce hydrogen peroxide and antimicrobial peptides (i.e., bacteriocins), which allow them to maintain an advantage over the other microorganisms in competition for nutrients and niche [8,9]. To thrive in such a rough and often hostile environment in the plaque oral biofilm, oral microbes must be able to survive and cope with all different insults. The cell envelope is a major sensory interface that provides the first line of defense in bacteria against these threats. It is also directly involved in surface adherence, intercellular interactions, and biofilm formation. Therefore, ensuring envelope integrity is crucial for *S. mutans* cells to survive and thrive in an environment.

*S. mutans* is known to possess strong capabilities to survive and adapt in response to various adverse conditions in the oral cavity, including low pH and oxidative stressors and nutrient deficiency [10]. Under certain conditions such as continuous consumption of sucrose and other readily metabolizable sugar rich diet and poor oral hygiene, *S. mutans* becomes numerically significant and leads to the demineralization of the tooth enamel and development of carious lesions [1,11]. Multiple biochemical pathways are employed by *S. mutans* to regulate its capacity to cope with environmental stressors such as acid and cell envelope stresses induced by antimicrobial agents [1]. They include two-component signal transduction systems such as LiaSR, molecular chaperones DnaK and GroEL, and BrpA (for biofilm regulatory protein A) [12,13,14,15,16,17]. *S. mutans* lacking LiaSR displayed increased susceptibility to lipid II cycle-interfering antibiotics and chemicals that perturb cell membrane integrity.

SMU.243 is part of the gene cluster that includes in its downstream, *mecA* and *rgpG* (Appendix A) genes known to play a critical role in stress tolerance responses and cell envelope biogenesis [18]. The product of *rgpG*, a member of the BrpA regulon [16], is an enzyme that catalyzes the first step of the biosynthesis pathway for rhamnosus-containing glucose polymers (RGP) [19]. *S. mutans* deficient of RgpG has no apparent change in growth rate under the conditions studied, but the deficient mutant does show severe defects in cell division and alterations in cell morphology [18,20]. The *mecA* gene encodes adaptor protein of the caseinolytic protease ClpCP machinery, whose deficiency was recently shown to lead to severe defects in cell envelope biogenesis and cell division and alteration of cell morphology, although there is also evidence that MecA can function independently of the ClpCP protease in regulation of *S. mutans* pathophysiology [21]. Located immediately downstream, SMU.244 encodes a homologue of undecaprenyl pyrophosphate phosphatase (UppP), which dephosphorylates undecaprenyl pyrophosphate (UPP), a precursor of essential carrier lipid undecaprenyl phosphate (UP) [22]. *S. mutans* with deletion of the *uppP* locus were shown to have increased sensitivity to cell wall antimicrobials such as bacitracin [23].

SMU.243 was predicted to encode a membrane-associated protein of 633 amino acid residues. By BLAST search of the NCBI database, the translated product was found to share homology to superfamily DUF2207 of bacterial proteins, a domain of unknown function but highly conserved in almost all bacteria. The aim of this study was to understand the impact of SMU.243 on the physiology of *S. mutans*. A SMU.243 deficient mutant was constructed by allelic exchange, and characterization of the resulting mutant showed that deficiency of SMU.243 in *S. mutans* caused major defects in stress tolerance responses and biofilm formation.

## 2. Methods and Materials

### 2.1. Bacterial Strains and Cultivation

*S. mutans* wild-type UA159 and its derivatives (Table 1) were maintained and grown in brain heart infusion (BHI, Difco Laboratories, Detroit, MI, USA) at 37 °C in an aerobic incubator with 5% CO_2_, unless otherwise specifically stated. When necessary, antibiotic spectinomycin (Sp, Sigma-Aldrich, St. Louis, MO, USA), erythromycin (Erm, Sigma-Aldrich), and/or kanamycin (Kan, Sigma-Aldrich) were added to the growth medium at the level of 1 mg/mL, 10 µg/mL, and 1 mg/mL, respectively. For growth studies, BHI broth medium adjusted to pH 6.0 or supplemented with methyl viologen at the concentration of 12.5 mM were also used, and the optical density of the bacterial cultures were measured and recorded continuously using Bioscreen C (Growth Curves USA, Piscataway, NJ, USA) with or without a layer of sterile mineral oil [17]. For E. coli strains, Sp, Erm and Kan were used at 100 µg/mL, 300 µg/mL, and 40 µg/mL, respectively.

### 2.2. Construction of Mutants and Transcription Initiation Site Mapping

A SMU.243 deficient mutant was constructed by allelic exchange using the well-established PCR-Ligation-Transformation strategy [25,26]. Briefly, the flanking fragments up- and down-stream of SMU.243 were amplified by PCR using high fidelity DNA polymerase Q5 (New England Biolabs, Ipswich, MA, USA) with forward and reverse primers 243Fw and 243Rv for the upstream and the downstream fragment, respectively (Appendix A). Following proper restriction digestions, the two fragments were ligated with a non-polar spectinomycin resistance marker that was digested with compatible enzymes [26]. Then, the ligation mix was used to directly transform *S. mutans* UA159 with inclusion of competence stimulating peptide (CSP) similarly as described previously, and mutants deficient of SMU.243 were selected on BHI agar plates with supplement of proper antibiotics [26,27]. To verify the accuracy of the deletional mutation, selected mutants were analyzed by PCR using the flanking 243Cf forward and reverse primers and by Sanger sequencing using primers NPSF and NPSR (Appendix A).

The transcription start sites were mapped using the Template Switching RT Enzyme Mix (NEB #M0466) by following the 5′ RACE protocol. For total RNA, *S. mutans* UA159 were grown in BHI broth, harvested at mid-exponential phase (OD_600_ ≈ 0.3), and then immediately treated with RNA Protect (Qiagen, Inc., Germantown, MD, USA). RNAs were extracted using hot phenol and purified using RNAeasy kit (Qiagen, Inc., Germantown, MD, USA), and DNase I treatment including an In-Column DNase I treatment was used to clean up the residual genomic DNA. 1 µg total RNA was used in the template switching reverse transcription, and the resulting cDNA with inclusion of a universal sequence attached to its 3′ were amplified by PCR using high fidelity DNA polymerase Q5 (NEB Biolabs, Ipswich, MA, USA), and the transcription initiation site was determined by Sanger sequencing of the PCR amplicon using primers shown in Appendix A.

To construct a complement strain, the SMU.243-coding region plus its promoter region were PCR amplified using Q5 DNA polymerase with 243Cp forward and reverse primers, and then cloned into integration vector pBGK3(2) [18]. Following sequence confirmation by Sanger sequencing, the resulting construct was used to transform the allelic exchange mutant, and the complement strains, as a result of double crossover homologous recombination, were isolated on BHI agar plate containing spectinomycin and kanamycin.

For construction of the *uppP* deficient mutant, the 5′ and 3′ flanking fragments were amplified by PCR using the UppPFw 5′ and 3′ primers for the upstream region and primers UppPRv 5′ and 3′primers for the downstream region, respectively. The *uppP* gene was deleted and replaced with a non-polar spectinomycin resistance marker using similar strategies as described above. For the *uppP* mutant complementation, primers UppPCp forward and reverse were used to amplify the *uppP*-coding region and its putative promoter region [23], and the resulting amplicon was introduced into the chromosome of the *uppP* mutant in single copy via pBGK3(2) similarly as described above.

### 2.3. Biofilm Formation

To assess the impact of SMU.243 deletion on biofilm formation, modified biofilm medium (BM) with inclusion of glucose (20 mM, BMG), sucrose (20 mM, BMS), or glucose (18 mM) and sucrose (2 mM) (BMGS) were used [28]. *S. mutans* wild-type, UA159 and its SMU.243 deficient mutant, TW434, were grown in 96-well plate or on hydroxylapatite (HA) discs (Clarkson Chromatography Products, Inc., Williamsport, PA, USA) that were vertically deposited in 24 well plates for 24 and/or 48 h [27,29]. For biofilm grown in 96-well plate, the sessile populations were stained with 0.1% crystal violet and measured in absorbance at wavelength of 575 nm with a spectrophotometer. Biofilms grown in HA discs were analyzed by using a laser scanning confocal microscope (Olympus Fluoview BX61, Center Valley, PA, USA) similarly as described previously [17,30]. For confocal microscopic analysis, HA discs were briefly washed in phosphate-buffered saline (PBS), 20 mM pH 7.0 to remove loosely attached bacterial cells, and then stained using Live/Dead bacterial staining kit (Invitrogen, Waltham, MA, USA) for 30 min which confers live cell with green fluorescence and membrane-compromised and dead cells with red fluorescence. Biofilms were then dissected using a confocal laser scanning microscope with a 60× water immersion objective lens. Post-acquisition analysis was carried out using SLIDEBOOK 5.0 (Olympus) and COMSTAT 2.0 [31], and the average thickness, biovolume, and surface area of the biofilms were calculated and compared as detailed previously [30,31].

### 2.4. Stress Tolerance Assays

The ability of the SMU.243 deletion mutant to withstand acid and oxidative stressors was analyzed by comparing the viability of the mutant and its parent strain following their exposure to low pH and hydrogen peroxide as described previously [17]. Briefly, overnight cultures were transferred to fresh BHI medium by 1:100 dilution and allowed to grow until mid-exponential phase with optical density (OD_600nm_) ~ 0.4. Cells were then harvested by centrifugation at 3000× *g*, 4 °C for 10 min, and washed once with 0.1 M glycine buffer pH 7.0 via centrifugation. For acid tolerance assay, cells pellets were resuspended in 0.1 M glycine buffer at pH 2.8 and incubated for 30, 45 and 60 min. For oxidative stress tolerance assays, bacterial cells were resuspended in 0.1 M glycine buffer, pH 7 containing 0.2% hydrogen peroxide (Fisher Scientific, Hampton, NH, USA) for 90 and 120 min. At each time point, serial dilutions were made and spread on BHI agar plates, and the survival rate at each time point was calculated and compared.

### 2.5. Cell Envelope Antimicrobial Susceptibility Assays

To evaluate the effects of SMU.243 deficiency on cell envelope properties, the deficient mutant was examined for its resistance against cell envelope antimicrobial agents, including bacitracin (Sigma-Aldrich), vancomycin (Sigma-Aldrich), penicillin G (Sigma-Aldrich) and non-cell envelope antimicrobials SDS (Sigma-Aldrich), D-cycloserine (Sigma-Aldrich), and Nisin (Sigma-Aldrich) by following the procedure described by Bitoun et al. [16]. In brief, overnight cultures of *S. mutans* wild-type and the SMU.243 deletion mutant were transferred to fresh BHI medium and allowed to continue to grow until OD_600nm_ ≅ 0.4, when the actively growing cultures were diluted with fresh BHI in 1:1000. Then, aliquots of the diluted cultures were loaded to 96 well plates and mixed properly with antimicrobial agents with proper serial dilutions. The cultures with proper antimicrobial agents were incubated aerobically at 37 °C with 5% CO_2_ for two days. By the end, the optical density was measured and relative cell density percentages were calculated [16]. The minimal inhibitory concentration (MIC) was determined as the minimum antibiotic concentration that allowed no significant growth when compared with blank medium. For minimal bactericidal concentration (MBC), 20 µL cultures were taken from the MIC plates and spread on BHI agar plates, and MBC was defined as the lowest antibiotic concentration that had less than 20 colony growth on BHI plates [16,32].

### 2.6. Statistical Analysis

Student *t*-test was used to further analyze the results and a *p* value of <0.05 was considered as statistically significant.

## 3. Results

### 3.1. Sequence Analysis of the SMU.243 Locus and Its Flanking Region

Locus SMU.243 was annotated to code for a membrane-associated protein with a domain termed DUF2207 of unknown function. Interestingly, as was initially annotated [33], the *S. mutans* protein was about two hundred amino acid residues shorter in its N-terminus region than its homologues in other bacteria. In the initial processes of allelic exchange mutagenesis and construction of the resulting mutant complementation, repeated efforts failed to generate a strain that was able to fully restore the phenotypes to the parent strain. Then, PCR amplifications and Sanger sequencing were used to further examine the coding sequence and its flanking regions. Surprisingly, multiple errors including a major gap were identified in the 5′ region of this locus. When analyzed, the results showed that instead of 1278 bp as was originally annotated, the actual open reading frame turned out to be 1902 bp, and the translated polypeptide should be 633 amino acids instead of the initial 426 amino acids (Appendix A) (GenBank # MW715639.1).

The transcription initiation of SMU.243 was also determined by 5′ Rapid Amplification of cDNA Ends using the Template Switching RT Enzyme Mix (NEB #M0466). The results showed that under the conditions studied, SMU.243 transcription started at the adenine site, which is 29 nucleotides away from the translation initiation site ATG (Appendix A). Further analysis of the promoter region revealed the putative ribosomal binding site (RBS), an extended −10 sequence that is conserved among some Gram-positive bacteria, and a −35 sequence.

### 3.2. SMU.243-Deficiency Causes Major Defects in Growth

To start to understand the roles of SMU.243 in *S. mutans* physiology, an allelic exchange mutant was constructed with the majority of its coding sequence replaced by a non-polar spectinomycin resistance element. Initially, a SMU.243 mutant was made with deletion of nucleotides 400 to 1572, relative to the translational initiation site ATG. By comparing the growth curves of the parent strain, UA159, the SMU.243 deficient mutant, TW434 (DF) had no major differences in growth rate when grown in regular BHI broth (Figure 1A). However, when incubated in BHI medium with adjusted pH 6.0, the growth rates of both wild-type, UA159 and the SMU.243 mutant, TW434 (DF) were reduced significantly, but significantly more reduction was observed for the mutant strain than its parent strain with an average doubling time of 279.5 (±17.3) minutes for wild-type vs. 469.6 (±113.8) minutes for the mutant (*p* < 0.05) (Figure 1B). Significant reduction in cell density of the overnight cultures were also observed with the SMU.243 mutant, with an average OD_600nm_ of 0.701 (±0.2) for the wild-type vs. 0.507 (±0.2) for the mutant (*p* < 0.01). As expected, the complement strain TW434C (DFC) carrying a wild-type copy of the coding sequence plus its cognate promoter restored the growth phenotypes with no differences from the wild-type.

When incubated in BHI with inclusion of methyl viologen (at 12.5 mM, final concentration), which is a chemical commonly used to induces intracellular oxidative stresses by production of superoxide radical and hydrogen peroxide [34], the SMU.243 deficient mutant (DF) displayed an extended lag phase of >11 h vs. 7.5 h for its parent strain (UA159), a reduced doubling time of 270 (±25.0) minutes vs. 729.3 (±34.5) minutes for the parent strain (*p* < 0.001), and a reduced culture density of OD_600nm_ 0.593 (±0.12) vs. 0.805 (±0.20) for the parent strain (*p* < 0.01) (Figure 1C). The complementation of the deficient mutant with the wild-type coding sequence plus its cognate promoter region completely restored the phenotypes of the mutant to the wild-type. These results suggest that the deletion of SMU.243 leads to growth defects when grown under low pH and in the presence of oxidative stressors.

### 3.3. Deficiency of SMU.243 Significantly Weakens the Tolerance of the Deficient Mutant to Hydrogen Peroxide

Acid killing and hydrogen peroxide challenge assays were used to further examine the ability of the SMU.243 mutant to survive and adapt low pH and tolerate oxidative stressors. When acid tolerance was analyzed by incubating the bacterial cells in a glycine buffer of pH 2.8 for periods of 30, 45, and 60 min, the SMU.243 mutant, TW434 (DF) exhibited a reduction of survival rate by 0.59-log after 60 min (*p* < 0.01), as compared to the wild type, UA159 (Figure 2A). When challenged with hydrogen peroxide for tolerance to oxidative stressors, the mutant also showed a reduced survival rate of >1−log after 60 min, compared to the parent strain UA159 (*p* < 0.001) (Figure 2B). The results further suggest that SMU.243 deficiency results in a weakened tolerance of the resulting mutant to low pH and hydrogen peroxide which are common stressors in the plaque environment.

### 3.4. Mutant with Deletion of SMU.243 Showed no Major Difference in Susceptibility to Cell Wall Antimicrobials

To investigate if SMU.243 deletion affects the ability of *S. mutans* to withstand stresses induced by cell envelope antimicrobials, the MIC and MBC against lipid II, non-lipid II inhibitors and cell membrane disrupting agents were analyzed. The results showed that relative to the wild-type, the mutant lacking SMU.243 displayed no significant differences in the MIC and BMC against bacitracin and the other antimicrobials tested (*p* > 0.05).

### 3.5. SMU.243 Deficiency Led to Major Compromises in Biofilm Formation

To evaluate the impact of SMU.243-deficiency on biofilm formation, biofilms formed by mutant TW434 (DF) with SMU.243 deletion was compared with wild-type UA159 using 96-well plate assays and confocal microscopy. As shown in Figure 3, when grown on a polystyrene surface in 96-well plates in BM with 20 mM of glucose (BMG), *S. mutans* strains develop limited biofilms, but relative to UA159, mutant TW434 (DF) developed less biofilms, although such differences were not significant (*p >* 0.05). When grown in the presence of sucrose at 2 mM (in BMGS) and 20 mM (in BMS), robust biofilms were measured with the wild-type, UA159. However, in comparison to the wild-type, biofilm formation by the SMU.243 mutant, TW434 (DF) was reduced by >4.12-fold (*p* < 0.001) and 1.7-fold (*p* < 0.05) during growth in BMS and BMGS, respectively. No significant differences were measured between the complement strain, TW434C (DFC) and the wild-type.

When grown on HA discs and analyzed using confocal laser scanning microscope, the biofilms of wild-type UA159 were robust, especially when grown in the presence of sucrose and distributed more evenly on HA surfaces (Figure 4A). Similar to 96-well plates, the SMU.243 mutant (TW434) also formed biofilms on HA discs, but in comparison to UA159, the SMU.243 mutant formed biofilms with significant reductions in both the depth and biovolume. However, there were no major differences in proportion and distribution of the red fluorescent dead/compromised cells between the mutant and the wild-type biofilms. As compared to the wild-type, the average thickness of the mutant biofilm was reduced by 2.94- and 4.01-fold on average, when grown in BMS (*p <* 0.05) and BMGS (*p* < 0.05), respectively (Figure 4B). After 24 h, the wild-type accumulated biofilms with an average biovolume of 1.8 (±0.71) µm^3^/µm^2^ when grown in BMS, while mutant TW434 (DF) averaged only 0.38 (±0.18) µm^3^/µm^2^ (*p <* 0.05) (Figure 4C). When grown in BMGS, the wild-type grew an average of 1.24 (±0.19) µm^3^/µm^2^, while mutant TW434 (DF) had only 0.38 (±0.22) µm^3^/µm^2^ (*p <* 0.05). No major differences between the mutant and its parent strain were observed in the surface area.

### 3.6. Deficiency of UppP Resulted in Alteration in Cell and Colony Morphology

A part of the cell envelope biogenesis /homeostasis cluster, *uppP* (locus SMU.244) was recently shown to encode a protein that is involved in the recycling of the carrier lipid and thus plays an important role in cell envelope biosynthesis and bacitracin resistance [23]. Relative to the wild-type, UA159, the *uppP* mutant generated by transposon insertion had an increased susceptibility to bacitracin, among other antimicrobials tested. In addition, the *uppP* mutant also displayed major defects in biofilm formation when grown in the presence of bacitracin. In the present study, an SMU.244 deficient mutant was constructed using allelic exchange, and the resulting mutant, TW435 was shown to possess two different colony morphology on agar plates with one variant, TW435d forming rough and dry colonies such as the parent strain and the other variant, TW435w forming wet, mucoid colonies with a round and smooth surface (Figure 5). Unlike the “rough” variant, the “smooth” variant grew more slowly than the parent strain especially under aerobic conditions, although no significant differences in doubling time were measured between the two variants and their parent strain, UA159. In addition, the “smooth” mutant also tends to form aggregates during growth in broth medium, unlike the parent strain and the “rough” variant. Consistently under optical microscopy, the cells with smooth colonies appeared to be bigger in size than the parent strain and those forming rough colonies (Appendix A). Complementation of the “smooth” variant with a wild-type *uppP* along with its cognate promoter in the complement strain, TW435WC, led to the restoration of cell morphology and colony morphology similar to the wild-type (Appendix A) and similar growth characteristics as the wild-type.

Further analysis on antimicrobial resistance showed that the SMU.244 deficient mutants had a reduced resistance to bacitracin with a MIC of 3.14 µg/mL vs. 66.5 µg/mL for the parent strain, consistent with Jalal et al. [23], and there were no differences in antimicrobial resistance tested between the two variant mutants. No differences were observed in survival rate following acid killing assay at pH 2.8 and hydrogen peroxide challenge assays. Interestingly, when grown in 96-well plate on HA discs, the wet mutant, TW435w with a smooth colony morphology formed slightly less biofilms as compared to the parent strain, while there were no significant differences between the rough mutant, TW435d and the wild-type, which is consistent with the findings of Jalal et al. [23].

To determine the genetic determinants responsible for the “rough” and the “smooth” morphological variations between the two mutants, genomic DNAs were prepared from the two variant mutants and then sequenced using an MiSeq Illumina platform to a depth of 1000 reads (Novegen, Dubin 18, Ireland). Comparative genomics analysis between the two variants against the parent strain, UA159 as a reference, confirmed the deletion of the *uppP* locus and its replacement with a non-polar spectinomycin resistance element, but revealed no other major differences in the genetic makeup between the two variant mutants and their parent strain. What factors contributed to the differences in cell and colony morphology between the two variant mutants await further investigation.

## 4. Discussion

SMU.243 is predicted to encode a member of the highly conserved DUF2207 superfamily of proteins with no known functions. Characterization of an allelic exchange mutant showed that deficiency of SMU.243 in *S. mutans* had no major effects on growth in a medium of neutral pH, but significantly reduced its growth rate and culture density overnight in a medium of pH 6.0, reduced its tolerance to low pH and especially, oxidative stressors, and led to significant compromises in biofilm formation, when comparing to its parent strain. As the immediate upstream element, SMU.242 coding for another hypothetic protein is located in the opposite orientation, and the downstream *uppP* is transcribed under its own promoter [23], as was previously shown by Jalal et al. [23], the allelic exchange mutagenesis of SMU.243 with a non-polar antibiotic resistance marker should have little or no impact on the flanking regions. Consistently, complementation of the deficient mutant with the SMU.243-coding sequence plus its cognate promoter was able to fully restore the phenotypes analyzed. These results suggest that the phenotypes observed with the SMU.243 mutant can be attributed solely to the deletion of the SMU.243 gene and is distinct from the phenotypes observed with deletion of *uppP*.

Biofilm formation is known to be highly regulated, and multiple factors have been shown to play a role in the regulation of *S. mutans* biofilms formation [1]. While the underlying mechanisms await further investigation, the weakened acid and oxidative stress tolerance responses as a result of SMU.243-deletion will certainly have an impact on the ability of the deletion mutant to form biofilms. When grown in the presence of sucrose, *S. mutans* is known to significantly enhance its biofilm formation because of its ability to produce adhesive extracellular polymers with sucrose as the substrate [1,4]. This can also be partly attributed to the observation that more significant differences in biofilms were measured when the bacterial strains were grown in BM-sucrose, although what other factors are involved in the sucrose-mediated biofilm formation remains unknown.

The DUF2207 Superfamily of proteins are hypothetical proteins featured with a highly conserved transmembrane domain that can be identified in at least 1460 different bacterial species, including Escherichia coli, Clostridium paraputrificum, Bifidobacterium italicum, and spirochaetes and archaea (http://pfam-legacy.xfam.org/family/PF09972#tabview=tab1, accessed on 24 July 2023). When analyzed by Protter (http://ulo.github.io/Protter/, accessed on 3 May 2023), an open-source program for interactive integration and visualization of predicted protein sequence features, the majority of the DUF2207 superfamily of proteins are integral transmembrane proteins. Represented by *S. mutans*’ SMU.243, these proteins possess a large extracellular N-terminus and a large intercellular C-terminus with a stretch rich of glycines and serines, although differences exist including the number of transmembrane segments (Appendix A). As predicted by DeepFRI, a structure-based protein function prediction program (beta.deepfri.flatironinstitute.org/workspace/GWJ9GE), the DUF2207 family proteins are more likely transmembrane transporters (Appendix A). As highlighted by the phylogenetic analysis (Clustal Omega, a multiple Sequence Alignment tool provided by EBI, https://www.ebi.ac.uk/Tools/msa/clustalo/, accessed on 24 July 2023), the *S. mutans* protein showed the best positivity and identity with the homologues in other oral streptococci and the group A and B streptococci (Appendix A). In fact, the genetic structure of the flanking regions is also highly conserved among the oral streptococci and the group A & B streptococci (Appendix A). As in *S. mutans*, the gene is flanked in its downstream with a cluster of genes including *uppP*, *mecA* and *rgpG*, which have been recently shown to play prominent roles in *S. mutans* pathophysiology including cell envelope biogenesis and cariogenicity [18,20,21,23,35].

The *uppP* gene encodes a undecaprenyl pyrophosphate phosphatase (UppP) that dephosphorylates undecaprenyl pyrophosphate and thus is involved in recycling of carrier lipid, playing an important role in biogenesis of cell envelope including peptidoglycan, teichoic acid, and lipopolysaccharides [36]. Recent studies by Jalal et al. have shown that deficiency of UppP in *S. mutans* results in reduction of phosphatase activity, increases in susceptibility to bacitracin, attenuates virulence, and causes compromises in biofilm formation when growing in the presence of bacitracin [23]. Consistently, our results also showed the allelic exchange mutation of *uppP* in *S. mutans* leads to growth defects, especially when grown under aerobic conditions, increased susceptibility to bacitracin, higher tendency of forming aggregates, and alteration of colony morphology, which again indicate defects in cell envelope biogenesis. In *S. mutans*, the transcription of *uppP* has been shown to be directed under its own promoter [23], and as expected, complementation with *uppP* and its cognate promoter restored its phenotypes in colony morphology (Appendix A) and bacitracin susceptibility. These results suggest that it is the lack of *uppP* that causes the defects in growth and alterations in colony and cell morphology.

The phenomenon that allelic exchange mutation of a target gene leads to the generation of mutant variants with distinct colony morphology is not uncommon in *S. mutans* [28,37], although the exact underlying factors remain mostly unclear. Recently, Turner et al. reported that *S. mutans* strains deficient of CidB resulted in the development of two different colony phenotypes, one “Rough” and the other “Smooth”, and the loss of the TnSmu2 genomic island and its ~20 kb 3′ region can be attributed to the formation of “Smooth” colonies and other related phenotypes [37]. We used genome resequencing using an Illumina platform to examine the two *uppP* variants, and comparative genomic analyses of the two genomes and the parent strain confirmed the deletion of *uppP*, but no major differences in genetic makeup were identified between the two mutant variants. It is therefore, unlike the *cidB* mutants, the stability of the genomic makeup is not part of the factors that led to the observed variations between the *uppP* mutants.

Our recent studies have shown that located immediately downstream of *uppP*, both *mecA* and *rgpG* genes play an important role in cell envelope biogenesis and cell division [18,21]. RgpG is the first enzyme of the biosynthesis pathway of the rhamnose-containing glucose polymers, which are the major cell-wall associated polymers in *S. mutans* [18,19]. Interestingly, when the “smooth” variant of the *uppP* mutant was transformed with a construct that carries the *rgpG*-coding sequence plus its cognate promoter [18], the mutant phenotypes including the “smooth” colony and cell morphology and bacitracin susceptibility appeared similar to the wild-type. Theoretically, introduction of an extra-copy of *rgpG* along with its cognate promoter at the *gtfA* locus in the chromosome will likely result in an increased level of enzyme RgpG and consequently, RGP production. These results suggest that the deletion of *uppP* and the resulting deficiency in carrier lipid recycling can be compensated at least partly through increased RgpG expression and enhanced RGP production. These results also suggest that similar to E. coli and some other bacteria [38], UppP homologues and /or additional enzyme(s) that can convert the pyrophosphate form of the carrier lipid to the mono-phosphate form likely exist in *S. mutans*. In support of this notion, the undecaprenol kinase (UdpK) in B. subtilis and *S. mutans*, formerly diacylglycerol kinase (DagK) [39], was recently shown to be able to phosphorylate free undecaprenol, generating carrier lipid undecaprenol phosphate [40].

Like *uppP*, deficiency of *mecA* was shown to result in elevated susceptibility of the deficient mutant to bacitracin [21]. In addition, the *mecA* mutant also displays major alterations in cell and colony morphology. Interestingly, similar results were also obtained when a copy of the *mecA*-coding sequence plus its cognate promoter [21] was introduced via an integration vector into the chromosome at the *gtfA* locus of the “smooth” *uppP* mutant. In *S. mutans*, *mecA* can be co-transcribed with *rgpG* and is autoregulated [21]. It is possible that when an extra-copy of *mecA* was introduced, the level of RgpG in the *uppP* mutant will likely be altered. Consequently, more carrier lipid will be available for biosynthesis of essential peptidoglycan, allowing the bacterium to grow better.

In summary, our results have shown that predicted as a member of the highly conserved DEF2207 superfamily of proteins, SMU.243 in *S. mutans* plays an important role in bacterial growth, acid and oxidative stress tolerance responses, and biofilm formation. Current effort is being directed to elucidation of the mechanism how SMU.243 alone and/or in conjunction with other genes in the cluster functions in regulation of the bacterial physiology.

## Figures and Tables

**Figure 1 microorganisms-11-01982-f001:**
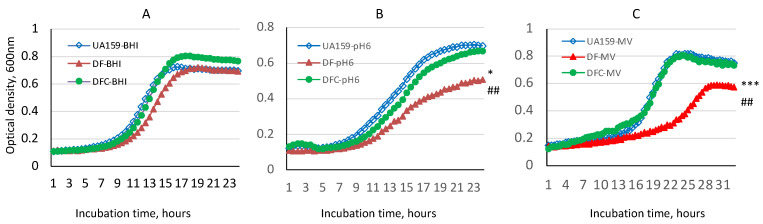
Growth study of the SMU.243 mutant. *S. mutans* UA159 (UA159), allelic exchange SMU.243 mutant (DF) and its complement strain (DFC) were grown in regular (**A**) BHI, (**B**) BHI adjusted to pH 6.0 and (**C**) BHI containing methyl viologen (MV) at 12.5 mM, and the culture optical density was monitored continuously using a Bioscreen C at 600 nm. Data show that relative to the wild-type, the mutant displayed a reduced growth rate and a reduced culture density overnight, when grown at pH 6.0 (**B**) and in the presence of MV (**C**). * and *** indicate significant differences in growth rate at *p* < 0.05 and 0.001, respectively; and ## indicate differences in culture density at *p* < 0.01, when the mutant (DF) was compared to the wild-type (UA159).

**Figure 2 microorganisms-11-01982-f002:**
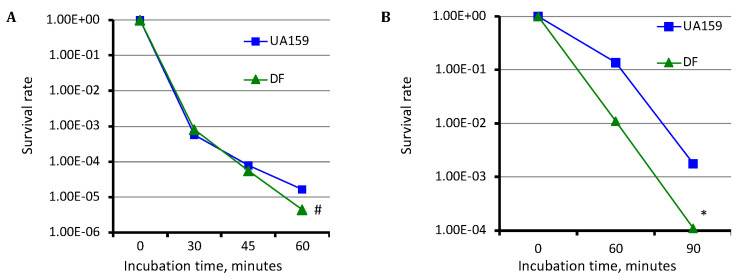
Hydrogen peroxide and acid killing assays. For acid killing (**A**) and hydrogen peroxide challenge (**B**) assays, *S. mutans* wild-type (UA159) and its SMU.243 mutant (DF) were grown in regular BHI broth and following acid killing at pH 2.8 and hydrogen peroxide challenge assay for indicated period of times, the survival rates of the mutant at each time point were compared to the parent strain and plotted. Data presented here are representatives of three separate sets of experiments. # and * indicates *p* < 0.01 and 0.001, respectively when the mutant (DF) was compared to the parent strain (UA159) under the conditions studied.

**Figure 3 microorganisms-11-01982-f003:**
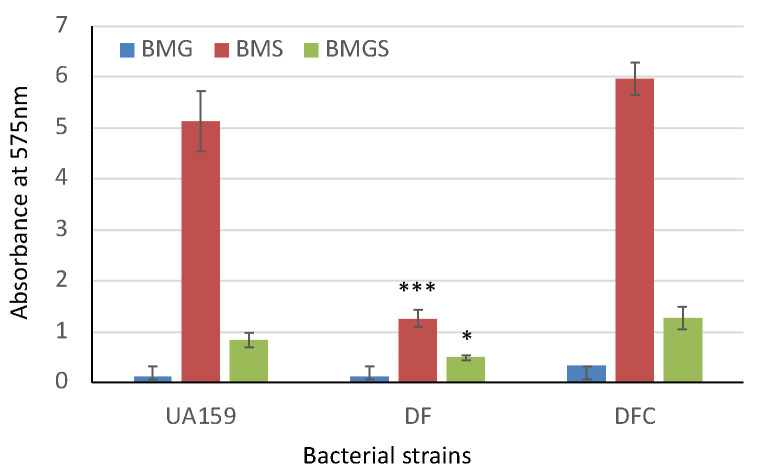
Biofilm formation by *S. mutans* wild-type (UA159), the SMU.243 mutant (DF), and its complement strain (DFC). Bacterial strains were grown in biofilm medium with glucose (BMG), sucrose (BMS), or glucose and sucrose (BMGS) on a polystyrene surface in 96 well plates and analyzed using crystal violet staining and spectrophotometry. Results represent mean absorbance (±standard deviation) at 575 nm from three independent experiments. Results showed major reductions in biofilm formation by the SMU.243 deletion mutant, especially when grown in the presence of sucrose. *, *** indicates statistical differences at *p* < 0.05 and 0.001, respectively, compared to its parent train under the same conditions.

**Figure 4 microorganisms-11-01982-f004:**
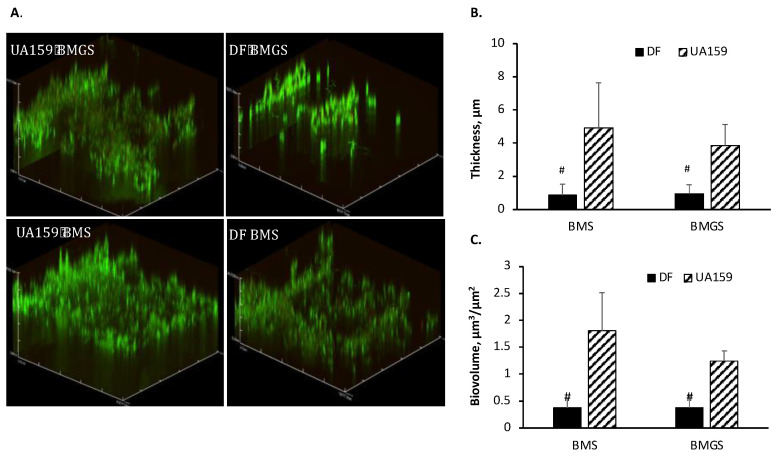
Confocal analysis of biofilms. *S. mutans* wild-type (UA159) and the SMU.243 mutant, TW434 (DF) were grown in biofilm medium with glucose (BMG), sucrose (BMS), or glucose and sucrose (BMGS), and biofilms on hydroxylapatite discs were analyzed using a confocal laser scanning microscope. Post-acquisition analyses were done using Comstat. Panel 4 (**A**) shows representatives of the reconstructed three-dimensional confocal images of biofilms of the wild-type, UA159 and its SMU.243 mutant (DF) when grown in medium with glucose and sucrose. Analysis of the biofilm (**B**) thickness in µm and (**C**) biovolume in µm^3^/µm^2^ of the mutant (DF) and its parent strain (UA159) was performed using Comstat with # indicating statistical differences at *p* < 0.05, when compared under the same conditions.

**Figure 5 microorganisms-11-01982-f005:**
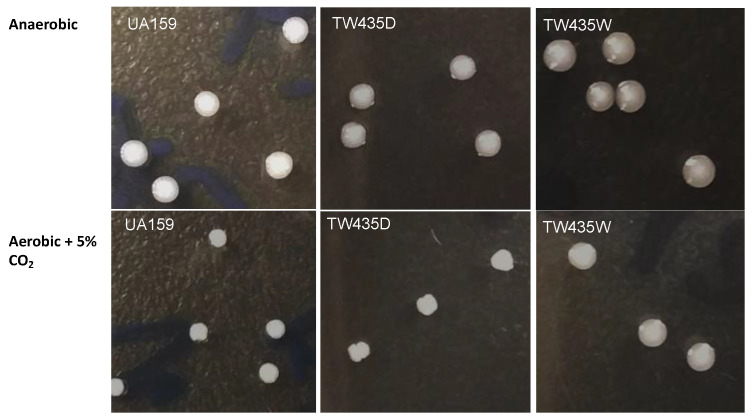
Colony morphology of the *uppP* mutant variants. *S. mutans* strains were grown on BHI agar plates in an anaerobic box or in an aerobic chamber with 5% CO_2_. Relative to the rough, dry colonies of wild-type (UA159), the *uppP* deficient mutant displayed “rough” (TW435d) and “smooth” (TW435w), two different colony morphologies, with the “rough” variant (TW435d) forming rough and dry colonies such as the parent strain and the “smooth” variant (TW435w) forming mucoid and wet colonies with a smooth surface. Images were taken using Samsung Galaxy Note V.

**Table 1 microorganisms-11-01982-t001:** Bacterial strains and plasmids used in this study.

Strains/Plasmids	Major Characteristics	References/Sources
*S. mutans* UA159	wild-type	ATCC
*S. mutans* TW434s	UA159/∆smu.243, Sp^r^	This study
*S. mutans* TW434k	UA159/∆smu.243, Kan^r^	This study
*S. mutans* TW434c	UA159/∆smu.243/*gtfA*::*Psmu.243*, Sp^r^, Kan^r^	This study
*S. mutans* TW435w	UA159/∆*uppP*, wet variant, Sp^r^	This study
*S. mutans* TW435d	UA159/∆*uppP*, dry variant, Sp^r^	This study
*S. mutans* TW435dc	UA159/∆*uppP*/*gtfA*::P*uppP*, Kan^r^, Spc^r^	This study
*S. mutans* TW435wc	UA159/∆*uppP*/*gtfA*::P*uppP*, Kan^r^, Spc^r^	This study
pBGK3(2)	Integration vector, Kan^r^	[18]
pDL278	Shuttle vector, Sp^r^	[24]
*E. coli* DH10B	Cloning host, *mcrA*, *mcrBC*, *mrr*, and *hsd*	Invitrogen, Inc.
pBGK3:P*sum.243*	pBGK3 with the *SMU.243* gene plus its promoter region	This study
pBGK3:P*uppP*	pBGK3 with the *uppP* gene plus its promoter region	This study

Note: Kan^r^, Erm^r^, and Sp^r^ for kanamycin, erythromycin, and spectinomycin resistance, respectively.

## Data Availability

The data presented in this study are available on request from the corresponding author.

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
