# Peer review of "Impacts of a DUF2207 Family Protein on Streptococcus mutans Stress Tolerance Responses and Biofilm Formation"

_microorganisms, 2023, doi:10.3390/microorganisms11081982_

Round 1

Reviewer 1 Report

The manuscript is a solid characterization of an open reading frame, SMU.243, encoded by the caries-associated S. mutans. The authors did identify a mistake in the original annotation of the orf and went on to characterize relevant phenotypes of a newly constructed mutant and the respective complemented strain, including stress tolerance and biofilm formation. The authors correctly conclude that the orf encodes for a protein from the DUF2207 family which is required for adequate stress response and biofilm formation, thus an important protein in the biology and virulence of S. mutans. The manuscript is well written, but contains some typos which can be corrected by re-reading the manuscript by the authors. In addition Fig. 4A has some issues due to text conversion.

Other issue:

line 48: better to say microbes instead of bacteria since C. albicans is definitely not a bacterium.

line 70: maybe reword the sentence. S. mutans is producing acid all the time, right now it reads as if they only start producing when they become numerically significant.

line 81: explain RGP biosynthesis

line 104: Why is S. mutans grown aerobically? This is especially important in the context of oxidative stress which is tested here and the possible accumulation of mutations in perR as recently reported (PMID: 33526613). Are the authors aware if the perR they have in their strain is wild type or mutated? It is possible that the observed phenotype might be much more pronounced if the here used strain carries a mutation in perR making the strain more tolerant against oxidative stress. This should be at least commented on.

please read carefully again, some minor typos are present in the manuscript.

Author Response

Responses to Reviewer #1:

The manuscript is a solid characterization of an open reading frame, SMU.243, encoded by the caries-associated S. mutans. The authors did identify a mistake in the original annotation of the orf and went on to characterize relevant phenotypes of a newly constructed mutant and the respective complemented strain, including stress tolerance and biofilm formation. The authors correctly conclude that the orf encodes for a protein from theDUF2207 family which is required for adequate stress response and biofilm formation, thus an important protein in the biology and virulence of S. mutans. The manuscript is well written, but contains some typos which can be corrected by re-reading the manuscript by the authors. In addition, Fig. 4A has some issues due to text conversion.

Response: Thank you. We have corrected all typos and fixed the issue with Fig. 4A.

Other issue:

line 48: better to say microbes instead of bacteria since C. albicans is definitely not a bacterium.

Response: The word “bacteria” is replaced with “microorganisms”, to be consistent elsewhere in the text.

line 70: maybe reword the sentence. S. mutans is producing acid all the time, right now it reads as if they only start producing when they become numerically significant.

Response: We have modified the sentence by deleting the words “produces acid that consequently”, as suggested.

line 81: explain RGP biosynthesis

Response: We have modified the statement to “biosynthesis pathway for rhamnosus-containing glucose polymers (RGP)”.

line 104: Why is S. mutans grown aerobically? This is especially important in the context of oxidative stress which is tested here and the possible accumulation of mutations in perR as recently reported (PMID: 33526613). Are the authors aware if the perR they have in their strain is wild type or mutated? It is possible that the observed phenotype might be much more pronounced if the here used strain carries a mutation in perR making the strain more tolerant against oxidative stress. This should be at least commented on.

Response: The conditions that were used to grow the S. mutans strains are mostly reflective of the environment on the tooth surface and are commonly used in study of the bacterium in the community. The strain we used in the study is the wild-type with no perR mutation.

Reviewer 2 Report

In this manuscript the authors describe the construction and characterization of a mutant in the SMU.243 open reading frame of Streptococcus mutans UA159 and compare the features of the SMU-243 mutant with the mutant of the downstream uppP gene. They found that the SMU-243 mutant was more susceptible to oxidative stress and low pH and showed reduced biofilm formation under specific growth conditions. In contrast, the uppP mutant displayed a major defect in cell envelope biogenesis and enhanced susceptibility to bacitracin. In addition, homologs of the SMU.243 protein which consists of the so-called DUF2207 domain almost over the entire length of the protein were identified and a phylogenetic relationship assessed and displayed.

General comments:

This is a well written and interesting manuscript characterizing the phenotypic consequences of a previously uncharacterized open reading frame. The major comment I have is that I am not entirely convinced about the ubiquitousness of the SMU.243 protein and its phylogenetic analysis. The authors need to provide evidence for this statement which is currently missing in the manuscript. In this context, how where the SMU.243 homologs chosen? To me some of the proteins seem to be only fragments of the entire SMU.243. Do all of these proteins which seem to be truncated versions of SMU.243 cover a similar region of the protein? An alignment is needed and needs to be shown in the manuscript. Equally the constructed tree does not show a quantitative assessment.

Other comments

Indicate in the abstract the identity of the Streptococcus mutans strain.

Why was complementation with the gene not shown for all experiments?

Figure 5: Colony morphology is not fully visible. Why not zoom in more?

Were the new genome sequences placed in a database?

Figure S1: Would be informative to add the promoter position and sequences.

                    Size of the respective gene: in bp?

Figure S2: Statement in the manuscript cannot be verified with this documentation.

Figure legends need to be extended in general.

is ok

Author Response

Responses to Reviewer #2:

In this manuscript the authors describe the construction and characterization of a mutant in the SMU.243 open reading frame of Streptococcus mutans UA159 and compare the features of the SMU-243 mutant with the mutant of the downstream uppP gene. They found that the SMU-243 mutant was more susceptible to oxidative stress and low pH and showed reduced biofilm formation under specific growth conditions. In contrast, the uppP mutant displayed a major defect in cell envelope biogenesis and enhanced susceptibility to bacitracin. In addition, homologs of the SMU.243 protein which consists of the so-called DUF2207 domain almost over the entire length of the protein were identified and a phylogenetic relationship assessed and displayed.

General comments:

This is a well written and interesting manuscript characterizing the phenotypic consequences of a previously uncharacterized open reading frame. The major comment I have is that I am not entirely convinced about the ubiquitousness of the SMU.243 protein and its phylogenetic analysis. The authors need to provide evidence for this statement which is currently missing in the manuscript. In this context, how where the SMU.243 homologs chosen? To me some of the proteins seem to be only fragments of the entire SMU.243. Do all of these proteins which seem to be truncated versions of SMU.243 cover a similar region of the protein? An alignment is needed and needs to be shown in the manuscript. Equally the constructed tree does not show a quantitative assessment.

Response: Information becomes available recently in the EMBL-EBI website (http://pfam-legacy.xfam.org/family/PF09972#tabview=tab1) that provides some details concerning the DUF2207 Superfamily of Proteins, their distribution among >1460 different bacterial species, sequence alignments, phylogenetic trees, and other related aspects. It is apparent that the DUF2207 family of proteins are wide-spread and diverse. While the majority of these proteins have one single DUF2207 domain, at least 213 proteins possess more than one DUF2207 domain. Represented by S. mutans SMU.243, the majority of the DUF2207 family proteins are integral transmembrane proteins, although differences exist including the number of transmembrane segments and the size between the different species. We have updated the phylogenetic tree analysis and added the alignment file of selected proteins, which highlight the close relationship of S. mutans SMU.243 with other major bacterial species.

Other comments

Indicate in the abstract the identity of the Streptococcus mutans strain.

Response: The strain was indicated in the abstract, as suggested.

Why was complementation with the gene not shown for all experiments?

Response: Complementation was done mostly to prove the concept. We agree that it would be ideal to have the complement strain included in all experiments, but it is also often difficult to do so due to time restraint and /or limits in access to certain resources. 

Figure 5: Colony morphology is not fully visible. Why not zoom in more?

Response: The colonies of the wild-type S. mutans were typically rough and dry under aerobic conditions. In comparison, the TW435W colonies were smooth and shining. We agree that they were kind of small. We chose the same settings for the images, so that proper comparisons could be made. 

Were the new genome sequences placed in a database?

Response: The corrected sequence information was submitted to NCBI database, and the GenBank number MW715639.1 is included in the manuscript.

Figure S1: Would be informative to add the promoter position and sequences.

Size of the respective gene: in bp?

Response: The sizes in basepair of the different genes were presented underneath the arrows, as indicated. The promoter region including the transcription initiation site was presented in Figure S2c. Considering the space limit with Figure S1, we elected to leave this information in Figure S2c. 

Figure S2: Statement in the manuscript cannot be verified with this documentation.

Response: It is apparent that under optic microscope, the uppP mutant with “Smooth” colony morphology appeared to be bigger than the wild-type and the other uppP mutant with “Rough” colony phenotype. This is obviously preliminary study. Confirmation will wait for electron microscopic analysis.

Figure legends need to be extended in general.

Response: Proper modifications have been made with the figure legends including those in the supplemental files.

Reviewer 3 Report

The authors have focused their works on a gene encoding a protein of unknown function in Streptococcus mutans, smu.2443, which is part of a cluster of genes all involved in envelope synthesis and consequently in stress adaptation and biofilm formation.

The context is clearly described in the introductory section, with the role of the bacterium in cariogenesis and the involvement of biofilm, and the need for the bacterium to tolerate the stresses encountered in the oral cavity.

 The description of smu.243's genomic environment provides a clear understanding of how the authors came to be interested in it and to seek to understand its role.

Methods are well adapted to answer to the questions and the results are robust. I think that a very interesting study but the authors did not discuss their results enough, or not in the good way.

My comments

Please add a little more on what is “RGP biosynthesis” line 81.

Line 182, cells are harvested at 4°C before challenged with an acid stress or an oxidative stress. Don’t you think that this temperature of centrifugation does not induce stress response by itself? and therefore affects subsequent stress responses?

Check that name of bacteria are in italics. Ex legend of figure 1 line 242, in figure 2 line 274.

Figure 4A is too little, can you increase the size? It should be easier to compare structure, since I think that you should describe them in the main text. The authors said line 169 in mat et met section that they used live/dead staining kit but the images of figure 4A is only green, is there any dead or damaged cells? the authors should discuss that.

In mat et met, the authors explain how they have mapped the transcription site “2.2 line 114. But the result of this seems to be explain only in one line (line 224). Since there is only one sentence, it’s not clear enough if this is the result of the mapping or just an observation by in sillico analysis. I suggest to add a figure showing a map and an analysis of the potential promoter sequence, is there any consensus sequence recognized by regulators like BrpA or involve in stress response.

I think that discussion is not enough focused on stress response whereas many results shown here are linked to stress, probably also linked to biofilm formation...

Is there any hypothesis why sucrose enhances differences in biofilm formation comparing to glucose in the mutant strain? any link with hyperosmotic stress?

Author Response

Responses to Reviewer #3:

The authors have focused their works on a gene encoding a protein of unknown function in Streptococcus mutans, smu.2443,which is part of a cluster of genes all involved in envelope synthesis and consequently in stress adaptation and biofilm formation.

The context is clearly described in the introductory section, with the role of the bacterium in cariogenesis and the involvement of biofilm, and the need for the bacterium to tolerate the stresses encountered in the oral cavity. The description of smu.243's genomic environment provides a clear understanding of how the authors came to be interested in it and to seek to understand its role. Methods are well adapted to answer to the questions and the results are robust. I think that a very interesting study but the authors did not discuss their results enough, or not in the good way.

My comments

Please add a little more on what is “RGP biosynthesis” line 81.

Response: Modification has been made to clarify RGP. See also response to Reviewer #1.

Line 182, cells are harvested at 4°C before challenged with an acid stress or an oxidative stress. Don’t you think that this temperature of centrifugation does not induce stress response by itself? And therefore affects subsequent stress responses?

Response: We appreciate that the low temperature may potentially affect the bacterial physiology and stress responses. However, the impact is expected to be minimal, as the time is short and under the temperature, the biological processes are at their minimal state. To minimize the biological processes was the logic behind the use of low temperature for centrifugation.

Check that name of bacteria are in italics. Ex legend of figure 1 line242, in figure 2 line 274.

Response: The name of the bacterium is fixed.

Figure 4A is too little, can you increase the size? It should be easier to compare structure, since I think that you should describe them in the main text. The authors said line 169 in mat et met section that they used live/dead staining kit but the images of figure 4A is only green, is there any dead or damaged cells? the authors should discuss that.

Response: The panel in 4A has been modified, and consequently, they are bigger in size. On Dead/Live staining, there were some dead /compromised cells in red, as can be seen now in the enlarged panel, but there were limited under the conditions studied, and no major differences were observed between the wild-type and the mutant. A line has been added in results as suggested.

In mat et met, the authors explain how they have mapped the transcription site “2.2 line 114. But the result of this seems to be explain only in one line (line 224). Since there is only one sentence, it’s not clear enough if this is the result of the mapping or just an observation by in sillico analysis. I suggest to add a figure showing a map and an analysis of the potential promoter sequence, is there any consensus sequence recognized by regulators like BrpA or involve in stress response.

Response: The transcription initiation site was determined by 5’RACE. A few lines have been added in the text, as suggested. The details of the promoter region are currently presented in Figure S2c along with the other sequence data, which we think is appropriate. If needed, we can certainly move this figure to the front. Currently, there is no further information as to the elements involved in stress response.

I think that discussion is not enough focused on stress response whereas many results shown here are linked to stress, probably also linked to biofilm formation... Is there any hypothesis why sucrose enhances differences in biofilm formation comparing to glucose in the mutant strain? Any link with hyperosmotic stress?

Response: S. mutans is known to form significantly more biofilms in the presence of sucrose because of its ability to use sucrose as a substrate to synthesize adhesive glucose polymers. The weakened ability to tolerate acid- and oxidative stresses is likely a major contributing factor to the reduced biofilm formation, and the differences in biofilms from the parent strain could be further increased when grown in the presence of sucrose. However, it awaits further investigation if any particular factors related to the sucrose-mediated biofilm formation are undermined in response to SMU.243 deficiency in the mutant. A paragraph on these related aspects has been added to the discussion. Hyperosmotic stress is not known to have any major impact on S. mutans biofilm formation.

Round 2

Reviewer 3 Report

Thanks for all modifications done. 

I have no more comments.

Author Response

Thank you very much for the fair and constructive review.